# In-Vitro Biocompatibility and Hemocompatibility Study of New PET Copolyesters Intended for Heart Assist Devices

**DOI:** 10.3390/polym12122857

**Published:** 2020-11-29

**Authors:** Maciej Gawlikowski, Miroslawa El Fray, Karolina Janiczak, Barbara Zawidlak-Węgrzyńska, Roman Kustosz

**Affiliations:** 1Faculty of Biomedical Engineering, Department of Biosensors and Processing of Biomedical Signals, Silesian University of Technology, Roosevelta 40, 41-800 Zabrze, Poland; 2Foundation of Cardiac Surgery Development, Artificial Heart Laboratory, Wolności 345a, 41-800 Zabrze, Poland; kjaniczak@frk.pl (K.J.); bzawidlak@frk.pl (B.Z.-W.); romankustosz@frk.pl (R.K.); 3Faculty of Chemical Technology and Engineering, Department of Polymer and Biomaterials Science, West Pomeranian University of Technology, Al. Piastów 45, 71-311 Szczecin, Poland; mirfray@zut.edu.pl

**Keywords:** ventricular assist devices, polyesters, hemocompatibility, biocompatibility, PET, copolyesters, biocompatibility, thrombogenicity, hemolysis

## Abstract

(1) Background: The evaluation of ventricular assist devices requires the usage of biocompatible and chemically stable materials. The commonly used polyurethanes are characterized by versatile properties making them well suited for heart prostheses applications, but simultaneously they show low stability in biological environments. (2) Methods: An innovative material-copolymer of poly(ethylene-terephthalate) and dimer linoleic acid—with controlled and reproducible physico-mechanical and biological properties was developed for medical applications. Biocompatibility (cytotoxicity, surface thrombogenicity, hemolysis, and biodegradation) were evaluated. All results were compared to medical grade polyurethane currently used in the extracorporeal heart prostheses. (3) Results: No cytotoxicity was observed and no significant decrease of cells density as well as no cells growth reduction was noticed. Thrombogenicity analysis showed that the investigated copolymers have the thrombogenicity potential similar to medical grade polyurethane. No hemolysis was observed (the hemolytic index was under 2% according to ASTM 756-00 standard). These new materials revealed excellent chemical stability in simulated body fluid during 180 days aging. (4) Conclusions: The biodegradation analysis showed no changes in chemical structure, molecular weight distribution, good thermal stability, and no changes in surface morphology. Investigated copolymers revealed excellent biocompatibility and great potential as materials for blood contacting devices.

## 1. Introduction

The progress of modern medicine in the treatment of heart failure would not be possible without the development of materials engineering. Ventricular assist devices (VAD) are commonly used for the treatment of patients suffering from end-stage heart failure. VADs are utilized as a bridge to heart transplant [1] or—in selected etiology of heart failure—as a bridge to heart recovery [2,3]. Nowadays, the most frequently used VADs’ type is the rotary blood pump; however, in some applications, it is better to utilize pulsating VAD [1]. Independently from the VAD type, its construction requires the application of advanced biomaterials of high biocompatibility, good physical and mechanical strength, and wear resistance.

An extracorporeal, pneumatic ventricular assist device, POLVAD (Figure 1a) was developed and implemented in Poland in 1995 for clinical use. Up until now, it has been applied in more than 330 patients (F.R.K. Intra Cordis, Zabrze, Poland) [2,3,4]. In 2012, an upgraded model of extracorporeal pulsating VAD, ReligaHeart EXT (Figure 1b) was developed, and, currently, it is under clinical trials [5].

The family of Polish pulsating VADs ReligaHeart (for adults as well as for children) is made up of a new generation of biocompatible polyurethanes (PU) BIOSPAN and BIONATE (DSM Biomedical, Exton, PA, USA) [5].

Polyurethanes are very versatile polymers due to their good biocompatibility; therefore, they are widely used in medicine, especially in cardiovascular applications. They are characterized by unique physical and chemical properties, such as good low-temperature flexibility, low level of leachable substances, excellent hydrolytic stability, an exceptional smooth surface, and translucency after processing (injection molding). The exceptional smooth surface allows for decreasing the intravascular coagulation risk—a significant side effect in heart prostheses application [1,6]. These properties result in high biocompatibility, allowing a wide application in medical devices for long-term contact with the circulating blood [7].

Despite the obvious advantages of PUs, their main drawback is poor biostability associated with long-term usage, caused by susceptibility to biological degradation [8]. The degradation of the polyurethanes occurs as a result of hydrolysis, oxidation, metal induced oxidation, environmental stress cracking, enzyme-assisted degradation, etc., and it leads to the loss of mechanical properties and material fracture [9]. The analyses have shown that these materials exhibit a high creep during long-term fatigue testing [10].

Therefore, there is a pressing need to develop innovative polymers with controlled and reproducible physico-mechanical and biological properties in order to overcome these drawbacks and to be utilized in pulsating VAD manufacturing. Recently, new copolymers of poly(ethylene terephthalate) and dimer linoleic acid (PET-DLA) have been developed [11] and tested in order to evaluate their cytotoxic and hemolytic properties [12]. Dimers of linoleic acid are nonlinear, hydrophobic molecules being used for the synthesis of various polymeric systems, including polyamides [13], polyesters [14], and various photocurable networks [15]. The appropriate selection of hard and soft segment in segmented copolyesters [16] has allowed for tailoring the material parameters such as hardness, melt flow index, wear resistance, and others. Suitable physical and chemical properties of new copolymers, especially long-term fatigue resistance and dimension stability, was motivation to investigate their properties for the construction of Polish VADs [17]. However, these materials contain crystallizable hard segments of ethylene terephthalate thus hindering transparency of these copolymers [18]. Therefore, we report here for the first time the modification of PET-DLA materials with D-glucitol, a sugar-derived diol known to impart the glass transition temperature of various polymers, including polyesters, thus improving their transparency. In this study, we explore the biocompatibility and hemocompatibility of PET-DLA copolymers modified with a sugar-derived alcohol. The thermal properties were assessed with differential scanning calorimetry (DSC). The cytotoxicity and material stability in the simulated body fluid (SBF) as well as thrombogenic and hemolytic properties were evaluated in detail.

## 2. Materials and Methods

Two copolymers of poly(ethylene terephthalate) (PET) and dimer linoleic acid (DLA) with variable PET hard segments content, namely 65% wt. and 70% wt., were synthesized and investigated. The synthesis procedure was identical to that already reported in [18], differing only at the polycondensation step where D-glucitol (0.2 mol) was incorporated along with DLA. Briefly, the first step of the reaction was transesterification carried out in the presence of ethylene glycol (GE, Sigma-Aldrich, Poznań, Poland) and dimethylene terephthalate (DMT, Elana, Toruń, Poland), at 180 °C, until 95% of methanol was collected. Before the polycondensation step, DLA (Pripol 1009, generously provided by Croda, Gouda, The Netherlands) and D-glucitol (Sigma Aldrich, Poznań, Poland) were added to the reaction mixture; then, the pressure was decreased to 0.2–0.4 hPa and the temperature increased to 260 °C. The second stage of polymerization was carried out at 260–265 °C and at the pressure of 0.2 mbar. The progress of polycondensation reaction was followed by controlling the power consumption of the stirrer. Once the constant value of power consumption by the stirrer was achieved, the material was extruded from the reactor into cold water, and material thread was pelletized. Polymer samples, abbreviated here as PET-DLA 65 and PET-DLA 70, in two forms of discs (14 mm in diameter and 1.5 mm thick intended for thrombogenicity examination as well as 8 mm in diameter and 2 mm thick intended for all other tests) were prepared using the BOY 35E injection molding machine (Dr. Boy GmbH & Co. KG, Neustadt, Germany). Before tests, the materials were sterilized by radiation with a 10 MeV electron beam generated in a linear electron accelerator Elektronika 10/10 at dose of 25 kGy. Commercial medical grade polyurethane (PU) BIONATE (DSM Biomedical, Exton, PA, USA) was used as a reference material for thrombogenic properties assessment.

### 2.1. Differential Scanning Calorimetry (DSC)

The thermal properties of the novel copolymers were determined using differential scanning calorimetry (DSC) (Q100, TA Instruments apparatus, New Castle, DE, USA). Material samples, weighing between 10 mg and 20 mg, were first dried under vacuum at 60 °C for 24 h. The DSC measurements were carried out in a triple cycle, “heating–cooling–heating,” over a temperature range from −90 °C to 300 °C. The heating/cooling rate was 10 °C/min. The glass transition temperature (T_g_) was determined from the second heat cycle as the midpoint of the transition. The melting (T_m_) and crystallization (T_c_) temperature were determined as the temperature values corresponding to the maximum of endothermic curve and the minimum of exothermic curve, respectively.

### 2.2. In Vitro Cytotoxicity

The in vitro cytotoxicity was assessed according to ISO 10993-5 regulation on L929 fibroblasts incubated for 24 h in Medium 199 supplemented by 10% fetal calf serum (FCS). Live and necrotic cells were marked with fluorescein diacetate (FDA) and propidium iodide (PI), respectively. Cell morphology has been observed with the Axio Observer (Carl Zeiss AG, Oberkochen, Germany) microscope and AxioVision 4.6 software (Carl Zeiss AG, Oberkochen, Germany). Cytotoxicity evaluated with this method does not require testing on a reference material.

### 2.3. Thrombosis

The influence of PET-DLA on platelets activation was assessed using Impact-R apparatus (DiaMed GmbH, Cressier, Switzerland), by a method simulating the physiological shear stress in blood flowing through the medium diameter arteries wall [17]. Biomaterial samples (14 mm in diameter, 1.5 mm thick) were contacted with human blood under dynamic conditions, allowing the blood (130 mL) to move in a swirling motion over the biomaterial surface at a rotational speed of 720 RPM for 300 s. The reference material was commercial polyurethane based on polyesters BIONATE 90A (DSM Biomedical, Exton, PA, USA), which is often used in the production of clinical medical devices intended for long-term blood contact.

The platelet activation rate in the blood after contact with PET-DLA materials was evaluated, by labelling specific cell receptors of: CD61 for platelets, CD62P for active platelets with p-selectin expression, and CD45 for leucocytes. To assess the number of active platelets and leukocyte–platelet aggregates, a flow cytometer (Beckman Coulter FC500, Brea, CA, USA) was used. The platelet adhesion by number of cells adhered on the biomaterial surface was marked by a fluorescence microscopy. After the Impact-R test, the residue of red blood cells and other non-adhered cellular blood components were removed from a biomaterials surface, by washing it in PBS (phosphate-buffered saline) fluid. Then, the biomaterials were stained with primary antibodies: anti CD45 FITC and anti-CD62P PE. The incubation was carried out at a room temperature for 30 min. The inverted research microscope Axio Observer (Carl Zeiss AG, Oberkochen, Germany) with AxioVision 4.6 software (Carl Zeiss AG, Oberkochen, Germany) was used in order for specimen assessment.

Statistical analysis was as follows: the extreme values and outliers were identified using a Tukey’s test with a 1.5 factor. The non-parametric Kruskal–Wallis test (the equivalent of ANOVA) was used with a post-hoc test of multiple comparisons. When comparing two independent variables, the Mann–Whitney U test was used (which is the strongest, non-parametric alternative to the Student’s *t*-test). Nonparametric tests do not require assumptions about the random variable and/or homogeneity of the variance between groups.

### 2.4. Hemolysis Evaluation

Biomaterials’ samples (8 mm in diameter, 2 mm thick) were exposed to direct contact with 2 mL of blood in the stress-free conditions (hematological roller mixer, mixing time: 24 h. at temperature of 37 °C). The whole human blood anticoagulation was performed with CPDA-1 (Citrate Phosphate Dextrose Adenine Solution). BIONATE 90A polyurethane was the negative control. The blank was blood stored in a hematology tube and not in contact with other biomaterial.

The blood was qualified regarding hemolytic status on the basis of following parameters: total hemoglobin level and erythrocytes number—with the utilization of hematology analyzer (2800 BC VET, Mindray, Shenzhen, China) and plasma-free hemoglobin concentration—using hematology spectrophotometer (HemoCue Low HB, Radiometer, Copenhagen, Denmark). Blood at room temperature, not subjected to the experiment conditions, was used as a blind trial, and blood without contact with biomaterials subjected to the experiment conditions (slow mixing) was used as a negative control.

The effect of biomaterials on erythrocytes was determined by the control of the following parameters: free hemoglobin concentration increases in the platelet poor plasma fHGB [g/L], the selected morphological blood parameters (RBC—red blood cells, HGB—hemoglobin) and erythrocyte indicators (MCV—mean corpuscular volume, MCH—mean corpuscular hemoglobin, MCHC—mean corpuscular hemoglobin concentration). A biomaterial hemolytic index was calculated from the free hemoglobin concentration increase. The hemolysis degree was determined by hemolytic index calculations (H) in accordance with the ASTM F 756-00 guidelines.

### 2.5. Hydrolytic Stability

The hydrolytic stability of the materials in simulated body fluid (SBF—an acellular artificial physiological fluid with an ionic composition similar to human plasma) was carried out at 37 °C, with constant liquid stirring over the material samples (100 RPM centrifugation) for three different time periods: 30, 60, and 180 days. The ratio of materials sample weight to SBF volume was 1:10 (1 g:10 mL). After incubation, samples were washed with deionized water and dried at the temperature of 40 °C. The polymer structure was assessed with the Fourier transform infrared spectroscopy using 6700 Smart Snap Orbit ATR FTIR spectrometer (Nicolet, Thermo Fisher Scientific, Waltham, MA, USA) and OMNIC software (Thermo Fisher Scientific, Waltham, MA, USA). The polymer molecular weight was analyzed with the gel permeation chromatography (GPC), equipped with Viscotek pump VE 1122 (ViscoTec, Töging am Inn, Germany) and differential refractometer Shodex RI SE-61 (Showa Denko K. K., Tokyo, Japan), at a temperature of 35 °C. The thermo-gravimetric analysis has been performed with the TGA/DSC STARe System analyzer (Mettler-Toledo, Columbus, OH, USA), in the temperature range from 25 °C to 800 °C at a heating rate of 10 °C /min, with a constant flow of nitrogen 60 mL/min. Polymer surface morphology was assessed with scanning electron microscopy using an ESEM QUANTA 250FEG apparatus (FEI, Hillsboro, OR, USA).

## 3. Results

### 3.1. Phase Structure of PET-DLA Materials

The modification of PET-DLA copolymers was performed with the use of small amount (0.2 moll) of a sugar-derived alcohol, namely D-glucitol. The DSC thermograms (Figure 2) showed biphasic morphology typical for thermoplastic elastomers with low temperature glass transition (T_g_) of DLA soft segments and high temperature melting (T_m_) of PET hard segments. Such biphasic morphology is also typical for medical grade polyurethanes (including BIONATE used in this work as a reference material), thus imparting material excellent elasticity typical for rubbers and processing of thermoplasts. Importantly, this biphasic morphology is believed to be responsible for excellent blood-compatibility due to regulation of proteins aggregation and suppression of fibrin formation [19]. The melting point of copolymers is very similar for both materials due to very small difference in segmental composition: 65% wt. vs. 70% wt. hard segments, and thus 209.78 °C vs. 208.23 °C, respectively. Greater differences between both copolymers are seen at the glass transition region, where PET-DLA 65 shows T_g_ at 2.97 °C, while T_g_ for PET-DLA 70 has been found at 14.2 °C. Both copolymers show cold crystallization endotherms at 114.38 °C and 122.13 °C for PET-DLA 65 and PET-DLA 70, respectively.

### 3.2. In Vitro Cytotoxicity

The results of the experiment have been presented in Figure 3. No cell necrosis was observed. No significant decrease of cell density or no cell growth reduction was noticed. According to the cytotoxicity evaluation scale presented in ISO 10993-5, PET-DLA 65 and PET-DLA 70 were defined as non-cytotoxic materials.

### 3.3. Thrombogenicity

Results of statistical analysis of thrombogenicity assessment have been depicted in Figure 4, Figure 5, Figure 6 and Figure 7. As can be seen from Figure 4, no significant differences in platelets activity in the blood after contact with biomaterials were observed. The median of number of activated platelets (CD62P) as well as leukocyte-platelet aggregates (CD62P-CD45) were similar for tested PET-DLA 65 and PET-DLA 70 copolymers and the reference polymer BIONATE. The thrombogenicity test results showed that the PET-DLA materials were characterized similar to BIONATE 90A, a medical grade polyurethane currently used in the Polish extracorporeal cardiac prostheses, with a low number of activated platelets and aggregates, thus demonstrating excellent biocompatibility and expected low risk of intravascular coagulation. The similar results were achieved when biomaterials were evaluated after the contact with the flowing blood. The median of number of adhered platelets (CD62P) and leukocyte-platelet aggregates (AGR) was comparable for all tested materials (Figure 5).

### 3.4. Hemolysis Evaluation

The results carried out for the blank test and negative control indicate no adverse effect of the test conditions on erythrocytes. The initial blood morphology (RBC, HGB, MCV, MCH, MCHC) and total hemoglobin concentration were within the reference range. No significant changes of blood parameters were observed in the blood after contact with tested copolymers (Figure 6).

Hemolytic index was 0.14 ± 0.07% for PET-DLA 65 copolymer and 0.29 ± 0.07% for PET-DLA 70 material (Table 1). The tested biomaterials were classified as non-hemolytic (H < 2%) on the basis of the hemolysis range defined by the standard ASTM F 756-00: H = from 0% to 2%—hemolysis degree: non-hemolytic; H = from 2% to 5%—hemolysis degree: slightly hemolytic; H > 5%—hemolysis degree: hemolytic.

### 3.5. Hydrolytic Stability

The IR spectroscopy analysis of PET-DLA 65 and PET-DLA 70 materials before and after biodegradation tests (Figure 7) revealed no evidence of changes in chemical structure. Polymer samples were homogeneous, and there were no changes in FTIR spectra for tested copolymers after 30, 60, and 180 days of incubation in SBF. However—in a spectrogram of PET-DLA 65 after 180 days of biodegradation—the small amplitude harmonics at 3150–3200 cm^−1^ is noticeable. Harmonics located in this region of spectrogram originates from stretching oscillation of hydroxyl group which may indicate formation of ionic intermediates.

The molecular weight distributions marked by GPC before and after degradation were identical (Figure 8). For both biomaterials, the molar mass of specimens after 30 and 60 days of biodegradation was the same. However, a small decrease of molar mass was observed for PET-DLA 65 after 180 days of biodegradation (refer to Table 2).

TGA analysis results have shown no changes in decomposition temperature after 30, 60, and 180 days of degradation for both tested biomaterials (refer to Table 3 and Figure 9). The thermal decomposition of PET-DLA 65 started at T = 410 °C and ended at around T = 470 °C and for PET-DLA 70, the thermal decomposition started at T = 411 °C and ended as well at T = 470 °C, respectively.

A surface morphology of PET-DLA materials was smooth as observed with SEM analysis before and up to 180 days of degradation, without evidence of degradative damage being observed (Figure 10 and Figure 11).

All chemical tests of PET-DLA 70 and PET-DLA 65 revealed that biomaterials are chemically stable and resistant to biodegradation.

## 4. Discussion

Important prerequisites of every new material developed for a specific biomedical application are: good biocompatibility along with suitable physical, chemical, and mechanical properties and a high resistance to biological degradation. Recently, a new family of thermoplastic elastomers, segmented copolyesters of poly(ethylene terephthalate), and dimer linoleic acid (PET-DLA) were developed for heart assist devices [11,12,17]. Their properties can be tailored as these strongly depend from the hard and soft segments content, thus complex heart assist device parts can be manufactured from the same chemically identical but mechanically different materials. In order to fine-tune the materials’ properties, two polyester copolymers containing 65% and 70% of ethylene terephthalate hard segments modified with D-glucitol were investigated. Their thermal properties indicated typical thermoplastic elastomer behavior, similar to medical grade BIONATE polyurethane used for heart assist devices [11,19].

Despite the fact that the first work on blood compatibility of PUs dates back to the late 1960s and to the work of Boretos et al. [20], and the preliminary results of the studies were very promising for such applications as artificial heart [21], heart valves [22], and hemodialysis membranes [23], all these materials still show imperfect properties. Several studies on various polyurethanes have shown their good or satisfactory blood compatibility which is strongly related to their chemical composition and microphase separated structure composed of hard and soft segment domains [6,24,25,26,27,28]. The researchers synthesized the series of segmented polyurethanes using three different polyethers as soft segments, namely, poly(tetramethylene oxide) (PTMO), poly(propylene oxide) (PPO), and poly(ethylene oxide) (PEO), and two different diisocyanates, 2,4-toluene diisocyanate (2,4-TDI) and 4,4′ diphenylmethane diisocyanate (MDI), respectively. The study revealed that the platelet retention index showed lower values for PEO-PUs as compared to PTMO-PUs and PPO-PUs. The same results were observed in another study. Ma et al. [29] synthesized polyurethane (PU) materials with different contents of hard segments (20%, 25%, 30%). The polymers were prepared based on hexamethylene diisocyanate (HDI) and polycarbonate diols by solution polymerization. The results showed that the hemolysis of PU membranes were all less than 5%, with the lowest hemolysis index found for material with 25% hard segments. Moreover, cells cultured in the extracts from PU membranes containing a 25% hard segment proliferated as relatively more thriving, meaning that this composition of the material has the lowest cytotoxicity.

The novel polyurethane (PU)/mustard oil composites fabricated by an electrospinning technique have been characterized and investigated by Jaganathan et al. [30]. The researchers investigated the blood compatibility of the fabricated scaffold. The study showed that the hemolytic index value for the neat PU and fabricated composites was observed to be 2.73% and 1.15%, respectively, thus indicating that developed composites showed a non-hemolytic behavior signifying the safety of the composites with red blood cells contact.

Prasath et al. [31] investigated the blood compatibility and platelet adhesion of polyurethane surface modified with hydrochloric acid (HCl). The polyurethane, Pellethane^®^ 2102-90A, was treated in HCl for 30 min and 1 h. The results showed that the platelet adhesion on the sample surface of modified PU was less comparable to control samples. Moreover, the results of the hemolysis test showed that control samples revealed higher haemolysis than the modified samples.

In another study, new material constituted by a poly(ether)urethane (PEtU) and a silicone (polydimethylsiloxane (PDMS)) was evaluated [32]. The researchers investigated the cytotoxic effect of the fabricated porous membranes. Human umbilical vein endothelial cells (HUVECs) and a mouse fibroblasts cell line (L929) were cultivated with extracts obtained from materials containing 10, 40, and 100% (w/w) of PDMS. The commercially available Estane 5714-F1 and Cardiothane 51 were used as controls. Extracts were incubated up to 72 h with HUVECs and L929 cells. The cytotoxic effect was evaluated by light microscopy, cell viability (MTT reduction and neutral red uptake), and proliferation (5-bromo-2′-deoxyuridine incorporation) tests. The study showed no toxicity for each PDMS concentration investigated.

Activation of the coagulation system, together with the activation of platelets, is one of the most complex processes in the human body [33]. Due to its complexity, developing a material which completely eliminates the risk of clotting has not been possible so far. The contact of blood with an artificial surface results in starting a coagulation cascade, but these phenomena are of different intensity depending on the material, its surface structure, and blood contact: nature and time. Currently, there are lack of standard methods of biomaterials thrombogenicity assessment. The ISO 10993 standard contains some research recommendations regarding the leucocytes and platelets; however, the standard does not provide specific test methods. For this reason, custom methods have to be developed that impede comparing the results obtained in different laboratories.

All the examples clearly indicate that, despite numerous new PU systems being developed, no substantially improved (satisfactory enough to enter clinical studies) materials exist on the market.

From the point of view of the application of investigated PET-DLA biopolymers in cardiac surgery, one of the most important issue is to assure its low thrombogenicity. Due to the planned area of application (blood pumps, ventricular assist devices), the thrombogenicity of PET-DLA should be tested under shear stress conditions affecting the platelets. Therefore, a rapid test under dynamic conditions (called Impact-R) was used to verify how the biomaterials surface affects the platelets activation and adhesion [33]. Thus, we evaluated the platelets function in fresh human whole blood, flowing over the biomaterial surface in conditions similar to the physiological ones. The phenomena generated by the rotation of the cone over the tested surface corresponds to the laminar flow of arterial blood. The number of adhered and activated cells was assessed with relevant antibodies with the use of flow cytometry and fluorescence microscopy. It was not possible to find positive control; therefore, we tried to apply glass (which activates platelets due to zeta potential); however, all adhered material was rinsed during specimen washing. The negative control was commercial polyurethane Bionate (DSM, Biomedical, Exton, PA, USA), widely used in medical devices intended for long-term contact with blood. The study conducted showed that both investigated PET copolymers affect the coagulation system just like reference material in respect to platelets’ activation and its adhesion to the surface of biomaterials. There were no statistically significant differences between the number of activated platelets and platelet–leukocyte aggregates in blood in contact with PET copolymers and reference Bionate polyurethane. The ISO10993 standard does not define thrombogenicity assessment methods with precision; however, it recommends making a comparison to references with athrombogenic properties that are known based on clinical applications. On this account, the PET-DLA biomaterial investigated may be recognized as athrombogenic and suitable for application in blood-contacting medical devices.

Except for thrombogenicity, the next, fundamental aspect of the hemocompatybility evaluation was to assess the materials’ effect on erythrocytes. Erythrocyte hemolysis causes the release of intracellular hemoglobin into plasma, which is a serious threat to human health and life. Moreover, the plasma-free hemoglobin activates platelets [34], which may result in the formation of clots adhered to the surface of the material or appearing even in distant blood vessels. The assessment of the hemolytic properties included monitoring the changes in the number of erythrocytes and their parameters as well as an increase of plasma-free hemoglobin concentration. The negative control as well as a blind test allow for checking the correctness of biological experiments: it assures results independency of individual variability of blood. Hemolysis assessed based on the ASTM F 756-00 standard showed that both investigated PET-DLA 65 and PET-DLA 70 copolymers are non-hemolytic. Low hemolysis makes biomaterials suitable for application in medical devices intended for contact with blood.

Environmental degradation of polymers designed for heart prostheses is a very undesirable process as it affects the deterioration of physical and mechanical properties of the prosthesis and change of biological interactions with blood flowing through the heart prosthesis. To assess an environmental stability of new PET-DLA copolymers, a long-term (180 days) hydrolytic degradation study was performed in SBF. Advanced chemical analysis (FTIR, TGE, GPC, and SEM) did not reveal any significant differences between PET-DLA copolymers before and after biodegradation in SBF. Taking into consideration the interpretation of the ISO10993-9,13 standard, investigated materials may be recognized as chemically stable and resistant to biodegradation. Such a slight decrease of molecular weight can be associated with some leaching of short, oligomeric units from our condensation copolymer into degradation medium. However, considering excellent thermal stability and lack of changes in T_d_ and T_on_ accompanied with no signs of environmental degradation (cracks formation), we postulate that detected changes in molecular weight will not have an adverse effect on the overall performance of materials for extracorporeal heart assist devices. However, as we already stated, the long-term in vivo studies are needed to fully support the long-term stability hypothesis.

## 5. Conclusions

The newly developed copolymers of poly(ethylene terephthalate) and dimer linoleic acid (PET-DLA) modified with D-glucitol showed good biocompatibility in contact with L929 fibroblasts. These new materials were characterized by low thrombogenicity and hemolytic activity and have no negative effect on human erythrocytes. Thus, the PET-DLA copolymers are characterized by a similarly low thrombogenic activity as medical grade BIONATE polyurethane. PET-DLA materials revealed excellent stability in SBF during 180 days of ageing, thus showing great potential as materials for blood contacting devices. In order to confirm the complete material biocompatibility from the aspect of a possible heart prosthesis application, an additional detailed in vivo biocompatibility study in long-term contact with blood and tissues is required.

## Figures and Tables

**Figure 1 polymers-12-02857-f001:**
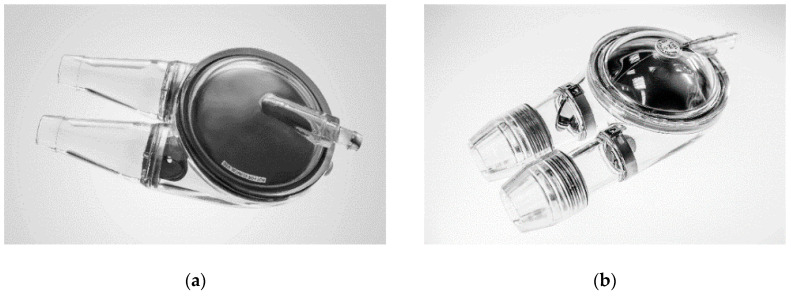
Polish, extracorporeal ventricular assist devices: (**a**) POLVAD and (**b**) ReligaHeart EXT.

**Figure 2 polymers-12-02857-f002:**
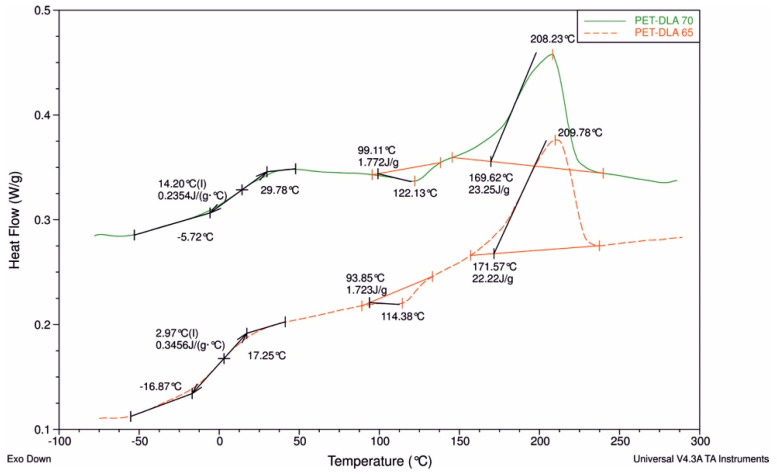
DSC thermograms of PET-DLA 65 and PET-DLA 70 copolymers (II heating run).

**Figure 3 polymers-12-02857-f003:**
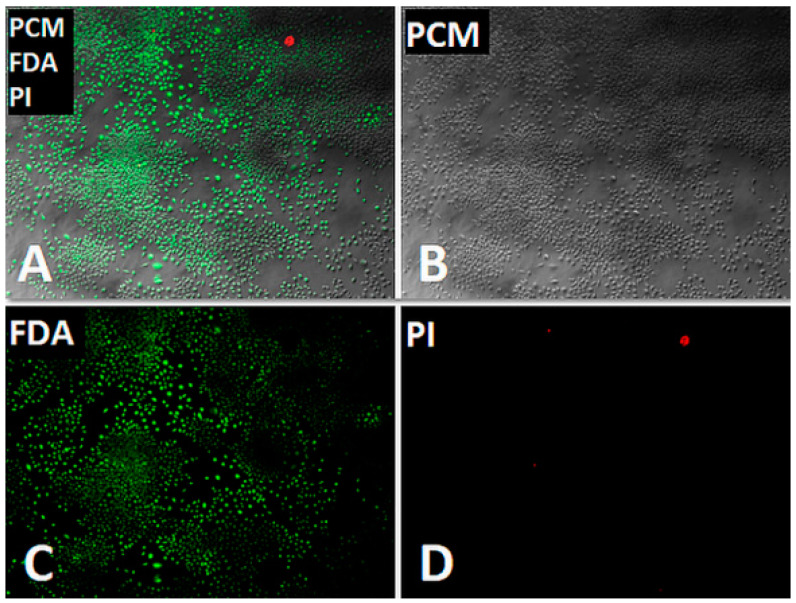
Microscopic cells evaluation in cytotoxicity test: (**A**) FDA + PI positive cells; (**B**) cells in transmitted light; (**C**) live cells toward FDA (green); (**D**) necrotic PI positive cells (red).

**Figure 4 polymers-12-02857-f004:**
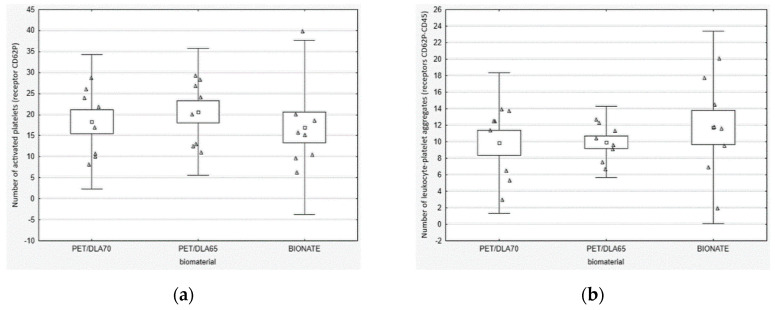
Activation thrombogenicity test: (**a**) platelets activation (receptor CD62P) in the blood after contact with biomaterials surface, (**b**) leukocyte-platelet aggregates generated in the blood by contact with biomaterials surface.

**Figure 5 polymers-12-02857-f005:**
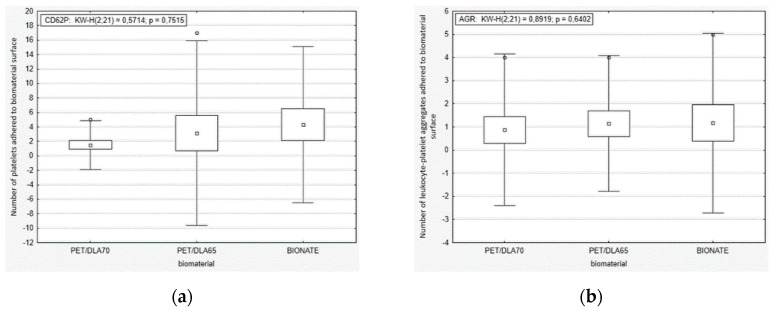
Adhesion thrombogenicity test: (**a**) activated platelets adhered to the biomaterials surface, (**b**) leukocyte–platelet aggregates adhered to the biomaterials surface.

**Figure 6 polymers-12-02857-f006:**
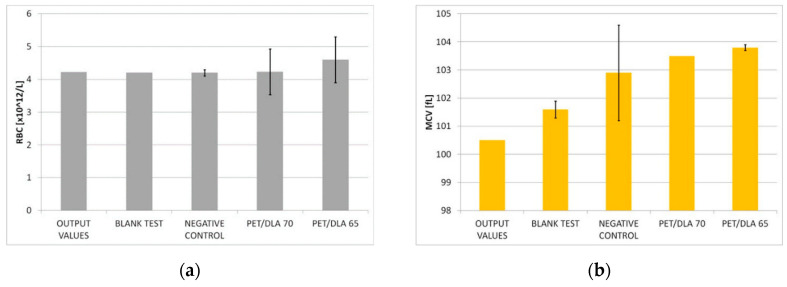
Blood parameters after contact with biomaterials surface: (**a**) RBC—number of red blood cells, (**b**) MCV—mean corpuscular volume, (**c**) MCH—mean corpuscular hemoglobin, (**d**) MCHC—mean corpuscular hemoglobin concentration, (**e**) HGB—total hemoglobin concentration.

**Figure 7 polymers-12-02857-f007:**
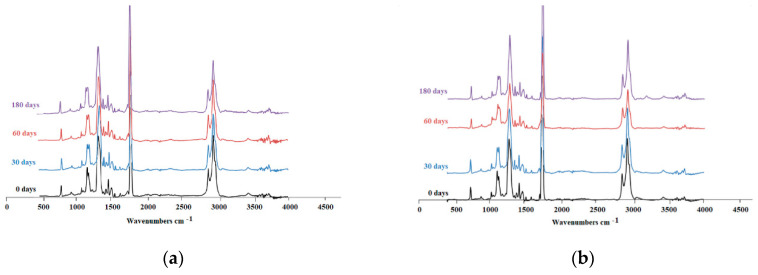
FTIR spectra for investigated copolymers at various biodegradation time intervals: (**a**) PET-DLA 70, (**b**) PET-DLA 65.

**Figure 8 polymers-12-02857-f008:**
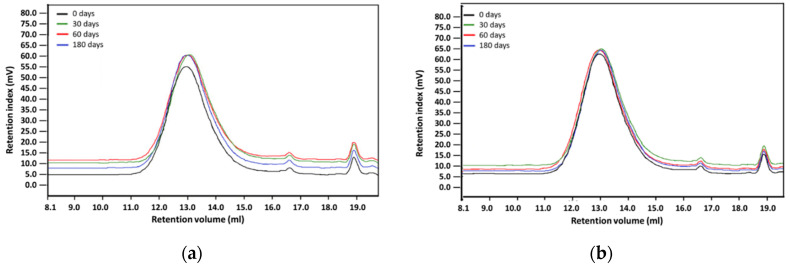
GPC chromatograms of PET-DLA copolymers at various biodegradation time intervals: (**a**) PET-DLA 70, (**b**) PET-DLA 65.

**Figure 9 polymers-12-02857-f009:**
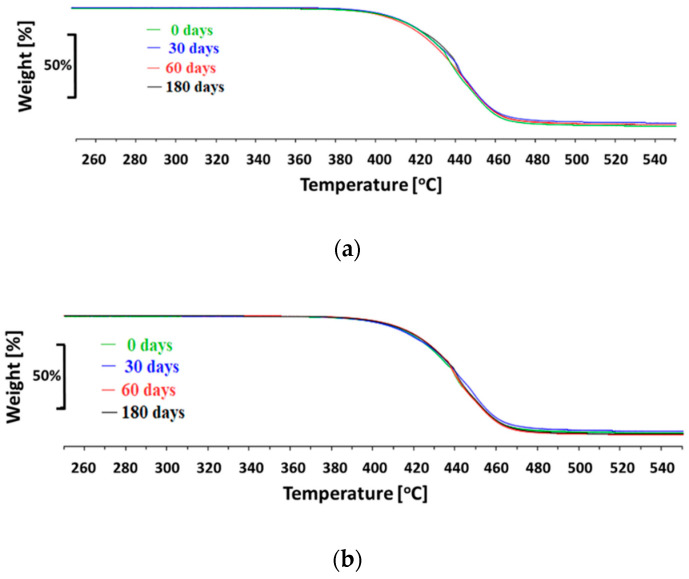
Comparison of TGA curves for copolymers after degradation in SBF: (**a**) PET-DLA 70, (**b**) PET-DLA 65.

**Figure 10 polymers-12-02857-f010:**
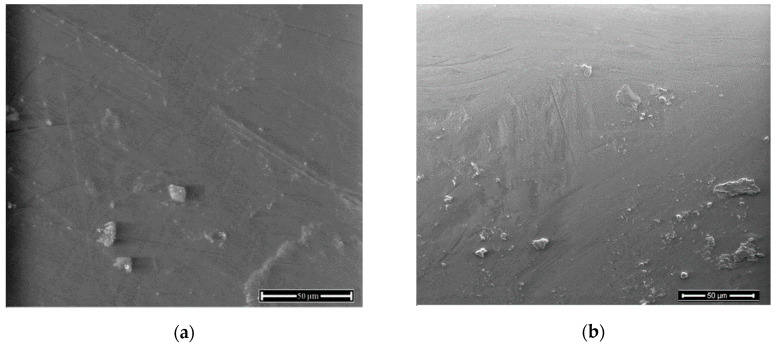
Surface of PET-DLA 65 copolymer: (**a**) before and (**b**) after biodegradation in SBF.

**Figure 11 polymers-12-02857-f011:**
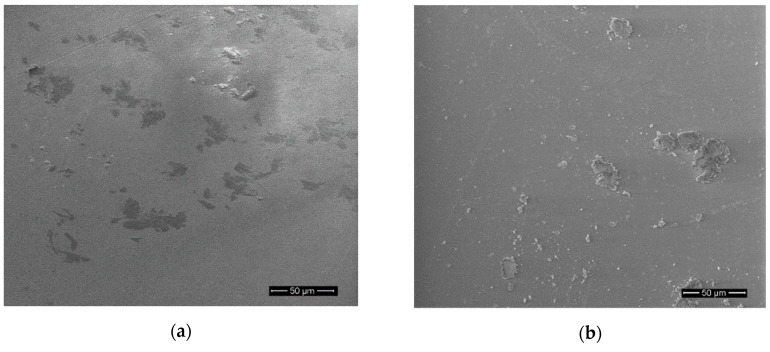
Surface of PET-DLA 70 copolymer: (**a**) before and (**b**) after biodegradation in SBF.

**Table 1 polymers-12-02857-t001:** Hemolytic parameters of examined biomaterials: hemolytic index and plasma-free hemoglobin.

Sample	Hemolytic Index [%]	Plasma Free Hemoglobin [g/L]
blank test	0.07	0.6
negative control	0.14 ± 0.06	0.7 ± 0.1
PET-DLA 70	0.28 ± 0.10	0.9 ± 0.1
PET-DLA 65	0.23 ± 0.06	0.7 ± 0.1

**Table 2 polymers-12-02857-t002:** GPC results of tested copolymers after degradation in the SBF.

Biomaterial	Duration [Days]	M_n_ [Da]	M_w_ [Da]	D_p_ [–]
PET-DLA 65	0	21,600	56,800	2.6
PET-DLA 65	30	20,600	52,700	2.6
PET-DLA 65	60	20,900	59,600	2.9
PET-DLA 65	180	18,700	53,000	2.8
PET-DLA 70	0	21,800	57,900	2.7
PET-DLA 70	30	20,000	54,400	2.7
PET-DLA 70	60	20,800	55,700	2.7
PET-DLA 70	180	19,800	51,100	2.6

Legend: Mn-number-average molecular weight; Mw-weight-average molecular weight; Dp-polydispersity, Pd = M_w_/M_n_.

**Table 3 polymers-12-02857-t003:** Thermal properties of tested copolymers after degradation in SBF.

Biomaterial	Duration [Days]	T_d_ [Da]	T_on_ [Da]	T_end_ [–]
PET-DLA 65	0	440	411	468
PET-DLA 65	30	449	410	473
PET-DLA 65	60	439	412	465
PET-DLA 65	180	441	413	467
PET-DLA 70	0	437	412	466
PET-DLA 70	30	442	413	474
PET-DLA 70	60	443	411	471
PET-DLA 70	180	440	413	471

Legend: T_d_-temperature of maximum weight rate; T_on_-onset temperature; T_end_-end temperature.

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
