# Peer review of "In-Vitro Biocompatibility and Hemocompatibility Study of New PET Copolyesters Intended for Heart Assist Devices"

_polymers, 2020, doi:10.3390/polym12122857_

Round 1

Reviewer 1 Report

Synthesized copolymers were used for medical application purposes. Though, PU is well known for this type of application. However, the proposed polymer also showed promising results and might be interested in this area.

Few points also need to be revised as follows:

  1. PU considered as a reference, they didn't consider PU during hemolysis and hydrolytic degradation analysis. Its also need to add these analysis.
  2. Please shortly explain the synthesis, you just mentioned the reference.
  3. The molecular weight decreased slightly, please explain the reason.   

Author Response

The authors would like to thank you for your insightful review. The authors agree with all the suggestions made by the reviewer. Detailed response to the rewiewer's comments can be found in the attached file.

Reviewer 2 Report

Review of the manuscript “In-vitro biocompatibility and hemocompatibility study of new PET copolyesters intended for heart assist devices” by Gawlikowski, El Fray, Janiczak, Zawidlak-Węgrzyńska, Kustosz.

In this paper, newly synthesized copolyesters of poly(ethylene terephthalate) and dimer linoleic acid (PET-DLA) are evaluated for its possible application for heart assist devices.  The in-vitro cytotoxicity, thrombogenicity under dynamic conditions and the hemolytic behavior of two copolymers with variable PET hard segments content (65% wt. and 70% w) are tested and compared with a medical grade polyurethane (BIONATE). The results show the new materials to be non-cytotoxic, athrombogenic and non-hemolytic. Therefore, the materials are suitable for applications in contact to blood. In addition, long-term (180 days) hydrolytic degradation in simulated body fluid has been analysed by FTIR, TG, GPC and SEM. As no significant differences has been observed in the samples before and after degradation, it can be concluded that these materials are resistant to biodegradation.

I think that the subject of the paper is of interest and the experimental work is good. I recommend it for publication in Polymers after some minor ammendement:

  • In figure 3 the correspondence between a, b, c and d figures and the legend should be revised.

Author Response

(The authors gave the same response as above.)

Round 2

Reviewer 1 Report

The manuscript revised accordingly. Its an interesting article and recommend for publication.